# Forced Degradation Studies and Development and Validation of HPLC-UV Method for the Analysis of Velpatasvir Copovidone Solid Dispersion

**DOI:** 10.3390/antibiotics11070897

**Published:** 2022-07-05

**Authors:** Bakht Zaman, Waseem Hassan, Adnan Khan, Ayesha Mushtaq, Nisar Ali, Muhammad Bilal, Dina A. Ahmed

**Affiliations:** 1Institute of Chemical Sciences, University of Peshawar, Peshawar 25120, Pakistan; zamanchem@gmail.com (B.Z.); waseem@uop.edu.pk (W.H.); 2Department of Pharmacy, Quaid-i-Azam University, Islamabad 45320, Pakistan; ishamalik1012@gmail.com; 3Key Laboratory of Regional Resource Exploitation and Medicinal Research, Faculty of Chemical Engineering, Huaiyin Institute of Technology, Huaian 223003, China; nisar.chemist.ali@gmail.com; 4School of Life Science and Food Engineering, Huaiyin Institute of Technology, Huaian 223003, China; bilaluaf@hotmail.com; 5Department of Pharmaceutical Chemistry, Faculty of Pharmacy, Future University in Egypt, New Cairo 1835, Egypt; dina.abbass@fue.edu.eg

**Keywords:** Velpatasvir, method development, forced degradation, recovery studies, degradation products, process impurities

## Abstract

Analytical methods for the drug substance and degradation products (DPs) are validated by performing forced degradation studies. Forced degradation studies of Velpatasvir (VEL) drug substance and Velpatasvir copovidone solid dispersion (VEL-CSD) were performed under the stressed alkaline, acidic, oxidative and thermal conditions according to ICH guidelines ICH Q1A (R2). VEL is labile to degrade in stressed alkaline, acidic, and oxidative conditions. It is also photolabile and degraded during photostability studies as described by ICH Q1B, and showed no degradation on exposure to extreme temperature when protected from light. A sensitive stability indicating HPLC-UV method was developed and validated for the separation of VEL and eight DPs. The DPs of VEL are separated using gradient elution of mobile phase containing 0.05% Trifluoroacetic acid (TFA) and methanol over symmetry analytical column C18 (250 mm × 4.6 mm, 5 µm) with a flow rate of 0.8 mL min^−1^. Simultaneous detection of all DPs and VEL was performed on UV detector at 305 nm. The performance parameters like precision, specificity and linearity of the method were validated using reference standards as prescribed by ICHQ2 (R1). Limits of quantification and limits of detection were determined from calibration curve using the expression 10δ/slope and 3δ/slope respectively. The proposed method is stability-indicating and effectively applied to the analysis of process impurities and DPs in VEL drug substance and VEL-CSD.

## 1. Introduction

Hepatitis C is considerable global health problem and the infected population of Hepatitis C Virus (HCV) is more than 170 to 200 million worldwide [1,2,3]. Velpatasvir (VEL) is a new molecule and a novel second-generation direct acting antiviral (DAA) for the inhibition of HCV. VEL has potent activity against all genotypes 1–6 of HCV and low toxicity EC50 values (6–130 pM). VEL 100 mg fixed dose combination tablets with Sofosbuvir 400 mg received marketing authorization from USFDA in June 2016 [4,5,6,7]. VEL is chemically carbamic acid with molecular formula C_49_H_54_N_8_O_8_ and structural formula shown in Figure 1A. It is a white to off-white, crystalline, non-hygroscopic, solid and belongs to BCS Class-IV having low pH dependent solubility and low permeability. To enhance the bioavailability of drug product, the pure drug of VEL is processed to copovidone solid dispersion and then used in pharmaceutical formulations. The bulk material of VEL is therefore supplied in the form of VEL-CSD containing VEL 50% *w*/*w* instead of pure dug substance [8,9]. Hence, the availability of stability indicating analytical method and forced degradation studies of VEL-CSD is the matter of concern.

The adverse effects of impurities and DPs due to manufacturing process, and uncontrolled storage conditions of intermediate and finished pharmaceutical products are crucial. There are strict guidelines to enhance the therapeutic effects and limit the adverse effects due to process impurities (IMPs) and DPs in pharmaceutical preparations [10,11]. The chemical stability of the drug substance is evaluated under the influence of various stressed environmental conditions to study the new molecules and its DPs. An efficient and validated stability indicating method is used to separate and quantify the impurities and DPs in the drug substance and pharmaceutical preparations [12,13].

Upon the literature survey, VEL has two major IMPs (Figure 1B,C) and eight DPs (Figure 2A–H) and no stability indicating HPLCUV method is available to quantify these degradation products in VEL-CSD. The official pharmacopeial monograph is also not available for the analysis of drug substance and pharmaceutical formulations. Very few analytical methods, focusing on simultaneous analysis of VEL and Sofosbuvir combined formulation, are reported in literature [13,14,15,16,17,18] Stability indicating methods for the analysis of VEL pure drug substance and degradation studies are also reported [19,20,21,22,23] but have limited information and discrepancies with published literature of FDA and MHRA [24,25]. As per MHRA public assessment report, VEL is photolabile, however it is reported stable in forced degradation studies published on UPLC [26]. The published methods exhibit certain disadvantages because of low sensitivity and are not applicable to the analysis of IMPs and degradation products in VEL-CSD. Similarly, the reported methods published for the analysis of DPs in VEL drug substance are not properly validated using reference standards of impurities and DPs. To the best of our knowledge, no study is reported till date to describe the forced degradation studies and validated stability indicating HPLC-UV method for the IMPs and DPs of VEL-CSD. For the purpose, we performed forced degradation studies of VEL-CSD as per the published guidelines of ICHQ1A (R2) [27,28]. Photostability testing [29] and all DPs were separated using simple and sensitive HPLC-UV method. The analytical method was accurately authenticated according to the guidelines ICHQ2 (R1) [30,31] using reference standards of IMPs and DPs. The proposed method is simple, precise and accurate and can easily be used for routine analysis in pharmaceutical testing and research laboratories.

## 2. Experimental

### 2.1. Materials and Chemicals

Reference standard of VEL 99.6% and VEL-CSD containing VEL 49.68% was provided by Anhui Yellen Pharmaceutical Co., Ltd., Hefei, China. Reference standards of IMPs i.e., IMP1 (C_47_H_52_N_8_O_6_) and IMP2 (C_39_H_45_N_7_O_5_), DPS i.e., DP1 (C_42_H_43_N_7_O_5_), DP2 (C_39_H_45_N_7_O_5_), DP3 (C_32_H_34_N_6_O_2_), DP4 (C_47_H_52_N_8_O_6_), DP5 (C_34_H_34_N_6_O_6_), DP6 (C_32_H_32_N_6_O_6_), DP7 (C_17_H_24_N_4_O_4_) and DP8 (C_16_H_20_N_2_O_4_) all purity < 84.0% was provided by Nantong Chanyoo Pharmatech Co. Ltd. (Nantong, China). Methanol HPLC grade, trifluoroacetic acid, sodium hydroxide, hydrochloric acid (37% *w*/*v*) and hydrogen peroxide solution (30% *w*/*v*) were purchased from Sigma-Aldrich (Darmstadt, Germany). Known excipients copovidone, croscarmellose sodium, microcrystalline cellulose, magnesium stearate, polyvinyl alcohol, titanium dioxide, polyethylene glycol and purified talc were provided by Genome Pharmaceuticals Pvt. Ltd. (Rawalpindi, Pakistan).

### 2.2. Instrumentation

Cecil low pressure quaternary gradient HPLC system comprised Adept CE-4104 pump, Adept CE 4200 variable wavelength UV detector, CE 4040 Solvent Degasser and CE4800-100 auto sampler by Cecil Instruments Limited, Peterborough, UK. The system was controlled by power stream chromatography manager version 4.2. Symmetry analytical column 250 mm × 4.6 mm, 5 µm, packing C18 (Waters, Milford, MA, USA) was used for analysis. Other equipment including UV-Visible spectrophotometer Shimadzu UV-2450, controlled by UV probe version 2.42, Climatic chamber for thermal and photo stability studies with florescent light of 1.2 milli lux h m^−2^ and UV light of 200 watt h m^−2^ (China), analytical balance AT-201 Mettler Toledo (Greifensee, Switzerland), ultrasonic bath SONOREX (Bandelin, Bandelin, Germany), Millipore vacuum filtration assembly and Milli-Q water distillation system (Millipore, Burlington, MA, USA) were also used in the studies.

### 2.3. Reference and Sample Stock Solutions

Stock solution of VEL reference standard 0.1 mg mL^−1^ was prepared by dissolving accurately 10.0 mg VEL in 100 mL methanol. The sample solution was prepared taking 20.3 mg VEL-CSD equivalent to 10 mg of VEL and dissolved in 100 mL methanol. The stock solutions 0.1 mg mL^−1^ of each impurity and DPs were also prepared in the same way taking equivalent quantity of 5 mg each and dissolved in 50 mL methanol separately. All the stock solutions were stored in refrigerator 2–8 °C in amber colored volumetric flask protected from light. The stock solutions were further diluted 10 µg mL^−1^ VEL and 0.01 µg mL^−1^ (0.1%) each impurity in method validation studies.

### 2.4. Forced Degradation Studies

Solutions of drug substance and copovidone solid dispersion equivalent to VEL 0.5 mg mL^−1^ subjected to hydrolysis in acidic and alkaline conditions using 5 M HCl and 1 M sodium hydroxide solutions [30]. Acidic, alkaline and neutral solutions were refluxed separately for 4 h and 8 h and the hydrolytic DPs were determined using the same validated procedure (Figure 3). The solutions of drug substance VEL and VEL-CSD having same concentration were kept in 10% H_2_O_2_ at room temperature and the extent of degradation were checked after 4 h and 8 h. Following ICH IB guidelines [28,29], VEL drug substance and VEL-CSD were exposed to 200 W h^m2^ UV light (320 to 400 nm) and 1.2 milli lux h m^−2^ visible light (400 to 800 nm) at 40 °C using photo stability chamber. The effect of heat and humidity on the drug substance was studied, exposing the VEL drug substance and VEL-CSD to dry heat 105 °C ± 5 °C and 80 °C/75 ± 5% RH for 24 h. The stability of solutions stored at room temperature (15–25 °C) and refrigerator (02–08 °C) for 7 days was also established, comparing the results with the results of reference standards’ freshly prepared solutions.

### 2.5. Optimization of Chromatographic Conditions

The chromatographic conditions for separation of impurities and DPs from VEL were optimized using gradient chromatographic system. First of all, the detection wavelength was studied on UV-Visible spectrophotometer and all the solutions 10 µg mL^−1^ were scanned from 200 nm to 400 nm. As the chemical structures of all impurities are closely similar to that of VEL, so the similar UV absorbance spectrum and absorption maxima 305 ± 3 nm were obtained (Figure 4). The optimum wavelength 305 nm was selected as suitable wavelength for simultaneous analysis of all DPs and IMPs. Now, the selection of the chromatographic conditions appropriate for separation of DPs and impurities having similar structure was a big challenge. The performance of different stationary phases i.e., Octyl Silica C8 (Waters), Octadecyl Silica C18 (Waters) and phenyl-hexyl (Accucore) columns was compared using same mobile phase with same gradient programs. The separation of composite reference solution containing all IMPs and DPs were studied using combination of different buffers ranging from pH 1.2 to 8.0 with methanol or acetonitrile as organic modifiers. Using C8 analytical column, the resolution of ingredients was quite difficult, and the impurities were not resolved from VEL at any pH described above. The resolution of all ingredients at acidic pH over C18 and phenyl-hexyl analytical columns was good, however the retention time of VEL and two DPs was more than 20 min. To achieve good resolution of each component in minimum run time, C18 analytical column was selected for further optimization. The gradient program for elution of acidic mobile phase A (0.1% TFA) with B (menthol) was adjusted to get optimal resolution in run time less than 15 min. Good separation was achieved (Figure 5 and Figure 6) using gradient elution: 0–2 min: 10% B, 2–6 min: 10–50% B, 6–12 min: 50–90% B, 12–13 min: 90–10% B, and 13–15 min: 10% B at a flow rate of 0.8 mL/min over C18 symmetry analytical column equilibrated at 30 °C. To validate the optimized conditions, all the performance parameters of the analytical method i.e., accuracy and recovery, precisions and repeatability, specificity and linearity were evaluated according to the prescribed limits and criteria of ICHQ1A (R2) guidelines.

## 3. Results and Discussion

### 3.1. Method Validation

Replicate injections (n = 5) of the reference composite solution containing VEL 10 µg mL^−1^ and impurities and degradation products each 0.1 µg mL^−1^ were analyzed on the optimized chromatographic conditions. Values for the system suitability parameters i.e., relative standard deviation (RSD), theoretical plates (N), symmetry factor (As) and retention factor (K_o_) were calculated from the resultant chromatograms. The results of RSD values were crosschecked against the acceptable limits for the drug substance (±2%) and impurities (±10.0%). All the peaks were symmetrical with the value of *As* near to 1.0 and the resolution of each component calculated with Relative Retention Time (*tRR*). The values theoretical plates for VEL were more than 3000 and *K_o_* was more than 2.0 (Table 1).

### 3.2. Recovery Studies

The recovery and accuracy were checked (n = 5) for six concentration levels ranging from 20 to 120% of the sample solutions of VEL 0.1 mg mL^−1^ (2.0, 4.0, 6.0, 8.0, 10.0 and 12.0 µg mL^−1^). To assure the detection and reporting threshold as prescribed by ICH guidelines Q3B (R2), each solution was spiked with the level of 0.05% of individual impurity and degradation product. VEL was recovered from the sample solution 98.0 ± 2% and the recovery for each impurity and DP was ±20% of the applied concentration (Table 2). The RSD value for recovery of VEL and each impurity was also within the acceptable limits ±2% and ±10% respectively.

### 3.3. Specificity

To evaluate the interference of other ingredients or excipients on the response and recovery of VEL and degrading products, the specificity of the method was checked. The recovery of VEL and degrading products was evaluated in the presence of excipients by adding an appropriate level of known excipients to the spiked composite solution. The response of the composite samples (n = 5) was compared with the response and individual reference solutions. It is evident from the chromatogram shown in (Figure 5) that each ingredient has its specific retention time, and there was no interference of other ingredients on the response of drug substance or any impurity.

**Figure 5 antibiotics-11-00897-f005:**
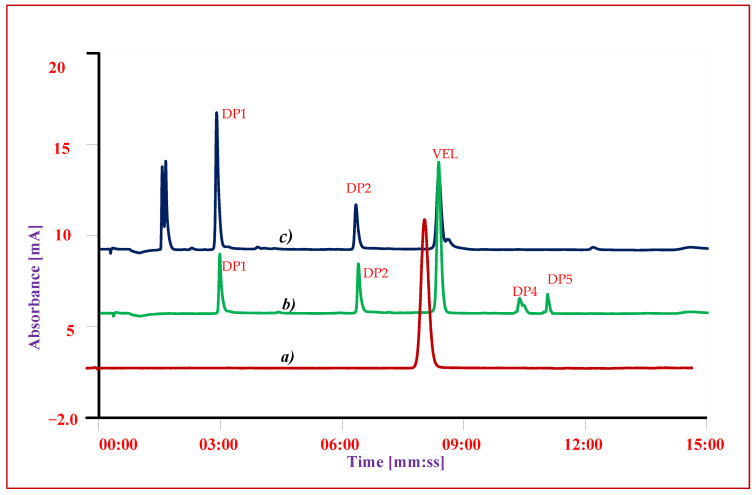
Chromatogram of VEL-CSD (**a**) Reference standard of VEL (**b**) Oxidative stress condition (**c**) Photolytic stressed studies.

### 3.4. Precision

To assure the repeatability of results, replicates (n = 6) of impurity spiked three concentration levels (low, medium and high) corresponding to 20, 60 and 120% of the sample solutions of VEL 0.1 mg mL^−1^ were analyzed. For repeatability, the RSD values of the results of same sample analyzed on the same equipment and same day were examined against the prescribed limits. The intraday precision was established, analyzing the same solution on different days and different equipment. For this purpose, the prepared sample solutions were provided to the pharmaceutical testing laboratory to analyze according to the optimized chromatographic conditions. The values of RSD for the responses i.e., peak area and retention time of VEL and impurities were less than 2.0% for the drug substance and 10.0% for all DPs respectively.

### 3.5. Robustness 

To demonstrate the effect of small changes in the chromatographic conditions on results, robustness studies were performed. The values of optimized chromatographic conditions were deliberately changed by ±2% and the consequence of these deviations were observed in the results of replicates (n = 6) of reference solutions. The results of the chromatographic response were checked against the values obtained using validated optimized chromatographic conditions. It was observed that the influence of the small changes in the chromatographic condition was negligible and the value of RSD for the peak area and retention time compared with standard values was less than 2.0%.

### 3.6. Linearity and Range

A linear relationship of the response and concentration is required to get a reportable range of an analytical method. Seven composite reference solutions varying from low concentration to high concentration, i.e., 10 to 120% of VEL 0.1 mg mL were prepared and analyzed on the optimized chromatographic conditions. The linearity of each individual component was examined using linear regression equation (A = slop C + Y intercept) from the graph of response plotted at on X axis verses concentration on Y axis. To establish the minimum detectable and quantifiable response limits (LOD and LOQ) for each analyte in the sample, the slope of the Calibration curve was determined. Based on the standard deviation of the response and slope, the values of LOD and LOQ shown in Table 3, were estimated using expression (3.3δ/slope) and (10δ/slope) respectively.

### 3.7. Forced Degradation Studies

Sample solutions after degradation studies were diluted to 0.1 mg mL^−1^ and analyzed using validated HPLC-UV method. The overall level of degradation was quantified by comparing the results of sample solution with freshly prepared reference solution (Table 4 and Figure 7). VEL both, pure drug substance and VEL-CSD, were hydrolyzed 16–20% in alkaline and acidic conditions and oxidized about 37–40% on exposure to oxidative stressed conditions. VEL is photolabile and the pure drug substance was degraded more than 15% during storage in photo-stability chamber for 7 days. VEL-CSD was also degraded during photolytic degradation studies, but the level of degradation was less than VEL pure drug i.e., 9.4% (Figure 6). The effect of dry heat and humidity on VEL and VEL-CSD was negligible and there was no degradation detected. Similarly, the solutions of VEL pure drug substance and VEL-CSD stored in amber color flasks at room temperature and refrigerator, were stable and showed no degradation during studies.

To identify and quantify the individual DP obtained in the studies, the results of sample solution were compared with results of composite reference solution. In overall studies DP1 (C_42_H_43_N_7_O_5_, Figure 2A), DP2 (C_39_H_45_N_7_O_5_, Figure 2B) and DP3 (C_32_H_34_N_6_O_2_, Figure 2C) were the major determined degradation product. DP1 and DP2 were quantified in both oxidative studies and photolytic studies; however, DP4 (C_47_H_52_N_8_O_6_, Figure 2D) and DP5 (C_34_H_34_N_6_O_6_, Figure 2E) were only detected in oxidative studies. DP3 was quantified in both alkaline and acidic conditions while DP6 (C_32_H_32_N_6_O_6_, Figure 2F) was detected only in alkaline and DP7 (C_17_H_24_N_4_O_4_, Figure 2G) and DP8 (C_16_H_20_N_2_O_4_, Figure 2H) in acidic condition. The results showed that VEL degraded to 4 DPs i.e., DP1, DP2, DP4 and DP5 in oxidative studies and into 2 DPs i.e., DP1 and DP2 during photolytic stress. On exposure to acidic stress, VEL hydrolyzed into 3 DPs i.e., DP3, DP7 and DP8 while in alkaline condition it degraded into 2 DPs i.e., DP3 and DP6.

**Figure 6 antibiotics-11-00897-f006:**
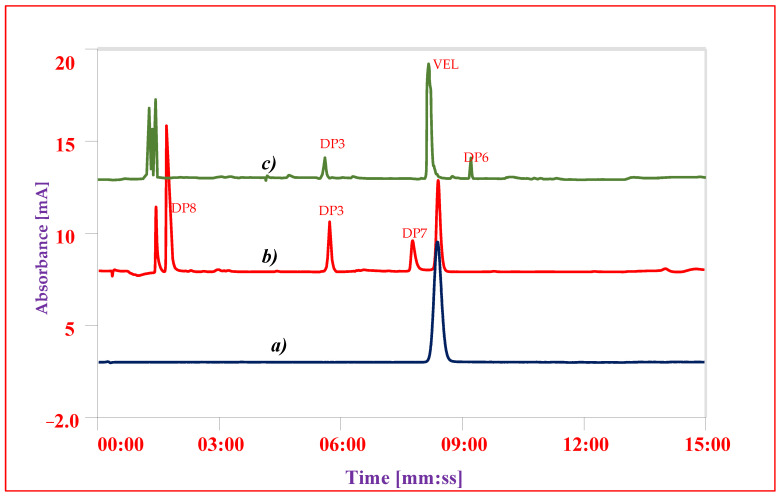
Chromatograms of VEL-CSD (**a**) Reference standard of VEL, (**b**) Hydrolysis in acidic condition (**c**) Hydrolysis in alkaline condition.

**Figure 7 antibiotics-11-00897-f007:**
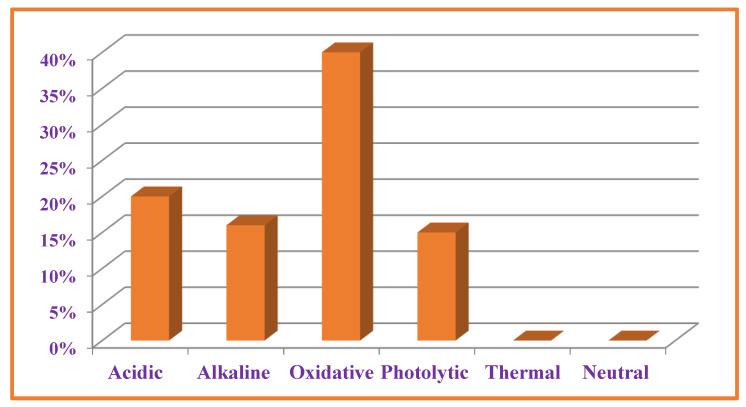
Degradation of VEL-CSD in different stressed conditions.

**Table 4 antibiotics-11-00897-t004:** Assay of VEL and VEL-CSD after exposure to stressed studies.

Medium	Material	At Refrigerator (02–08 °C) Protected from Light for 7 Days	At Room Temperature (15–25 °C) Protected from Light for 7 Days	At 80 °C Refluxed for 4 h Protected from Light	At 80 °C Refluxed for 8 h Protected from Light	At 40 °C) Exposed to 1.2 Milli lux h m^−2^ of Fluorescence Light for 7 Days	At 80 °C/75 ± 5% RH Protected from Light and Moisture for 24 h	At 105 °C Temperature Protected from Light and Moisture for 24 h
**In neutral condition**	VEL	100.13 ± 0.27 0.28%	98.14 ± 0.74 0.69%	97.51 ± 0.84 0.74%	98.27 ± 1.18 1.65%	-	-	-
	VEL-CSD	99.44 ± 0.79 0.65%	99.73 ± 1.14 0.79%	98.69 ± 1.17 1.20%	98.66 ± 1.12 1.24%	-	-	-
**Acid Hydrolysis** **(5M HCl)**	VEL	-	-	87.26 ± 3.24 2.47%	79.84 ± 4.46 5.12%	-	-	-
	VEL-CSD		-	90.41 ± 2.37 3.10%	94.56 ± 3.49 2.18%		-	
**Alkaline hydrolysis** **(1M sodium hydroxide solutions)**	VEL	-	-	90.51 ± 2.29 1.77%	87.98 ± 3.87 3.21%	-	-	-
	VEL-CSD	-	-	92.36 ± 2.24 1.96%	89.12 ± 1.14 1.25%	-	-	-
**Oxidative conditions** **(10% H_2_O_2)_**	VEL	-	-	76.66 ± 3.21 2.45%	79.64 ± 5.18 4.75%	-	-	-
	VEL-CSD			91.33 ± 6.41 4.82%	83.45 ± 4.79 2.87	-	-	-
**Photolytic condition**	VEL	-	-			84.16 ± 2.11 1.98%	-	-
	VEL-CSD	-	-			91.64 ± 1.75 154%	-	-
**Thermal stress**	VEL	-	-	99.54 ± 0.78 0.78%	98.71 ± 0.24 0.24%	98.27 ± 0.67 0.68%	99.10 ± 0.25 0.25%	99.16 ± 1.16 1.15%
	VEL-CSD		-	98.21 ± 01.31 1.67%	100.47 ± 1.61 1.52%	99.44 ± 0.75 0.38%	98.91 ± 0.31 0.46%	99.22 ± 0.78 0.80%

## 4. Conclusions

Stability indicating HPLC-UV method for the analysis of VEL-CSD was developed and validated using reference standards as per guidelines presented by ICH Q1A (R2). Forced degradation studies were performed on VEL drug substance and VEL-CSD. The reported IMPs and all DPs were separated and quantified using the same validated method. The specificity of the method for DPS and IMPs was conformed using reference standards from the manufacturer. The recovery of VEL in VEL-CSD and samples of stressed studies was determined and extent of degradation was evaluated accordingly. It was concluded from the results of validation studies that the developed method is specific, precise, and accurate for the intended purpose. HPLC-UV is readily available in all pharmaceutical research and testing laboratories. This validated method will help the researchers and analysts to assure the quality of novel and lifesaving antiviral drug VEL and VEL-CSD during routine quality control analysis.

## Figures and Tables

**Figure 1 antibiotics-11-00897-f001:**
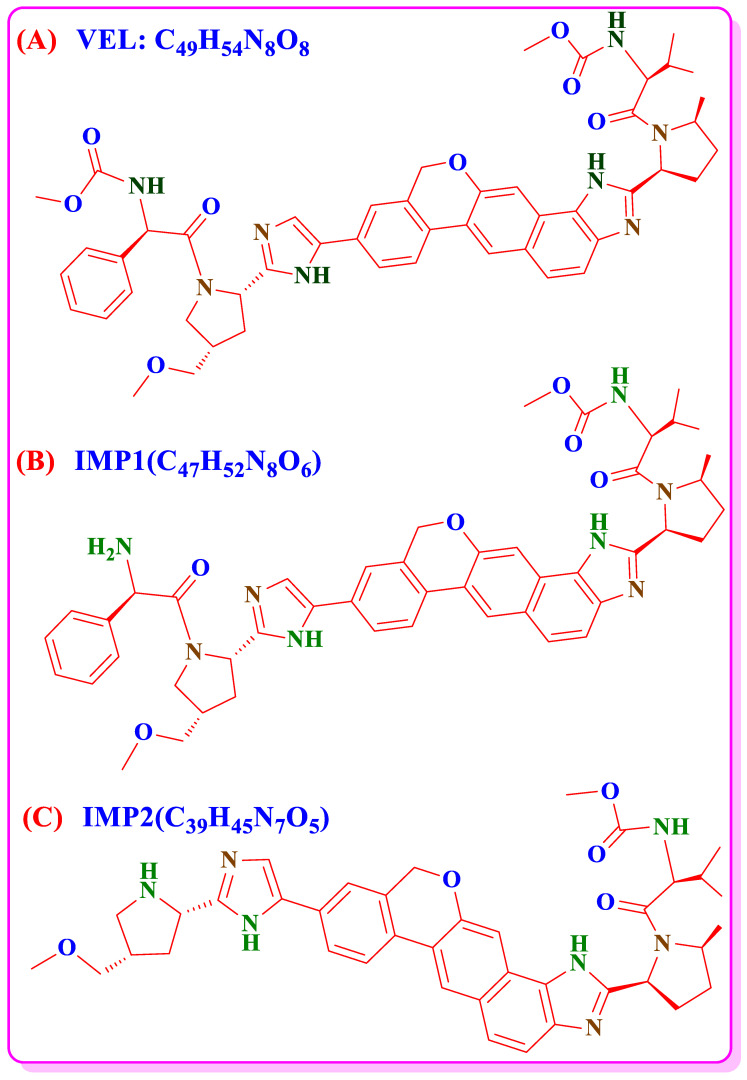
Chemical structures of (**A**) VEL, (**B**) process IMP-1 and (**C**) process IMP-2.

**Figure 2 antibiotics-11-00897-f002:**
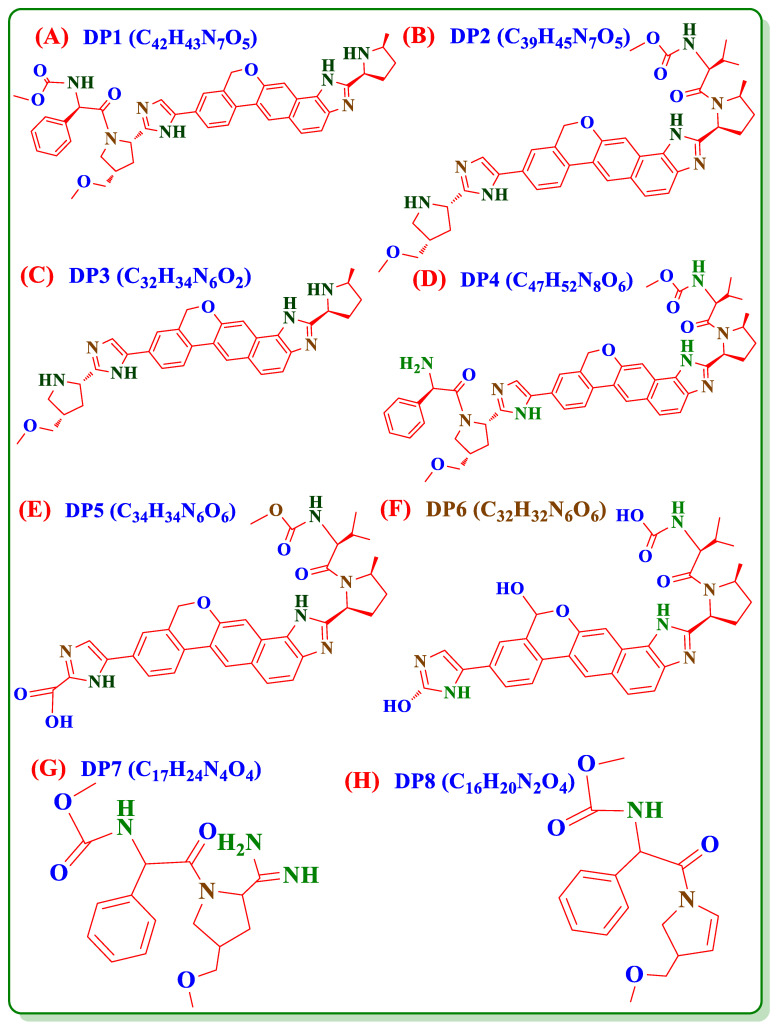
Chemical structures of degradation products of VEL (**A**) DP1, (**B**) DP2, (**C**) DP3, (**D**) DP4, (**E**) DP5, (**F**) DP6, (**G**) DP7 and (**H**) DP8.

**Figure 3 antibiotics-11-00897-f003:**
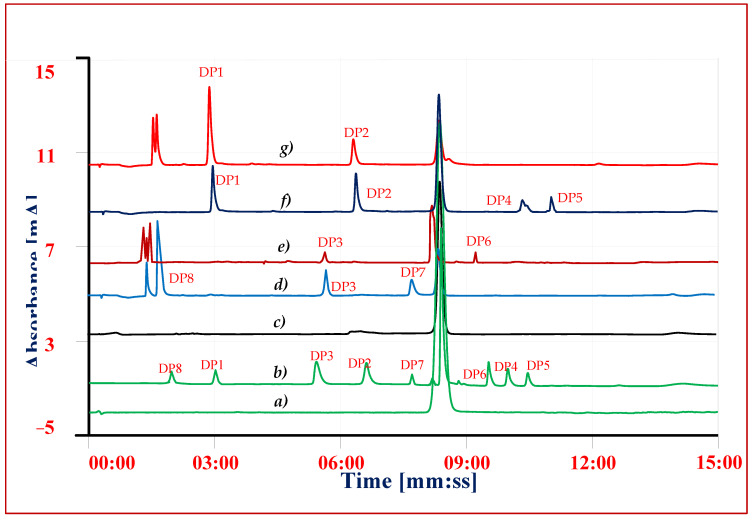
Chromatograms of forced degradation studies and reference standards (**a**) VEL reference standard (**b**) VEL composite reference standards, (**c**) VEL-CSD thermal study, (**d**) Acid degradation (**e**) Alkaline degradation, (**f**) Oxidative degradation (**g**) Photolytic degradation.

**Figure 4 antibiotics-11-00897-f004:**
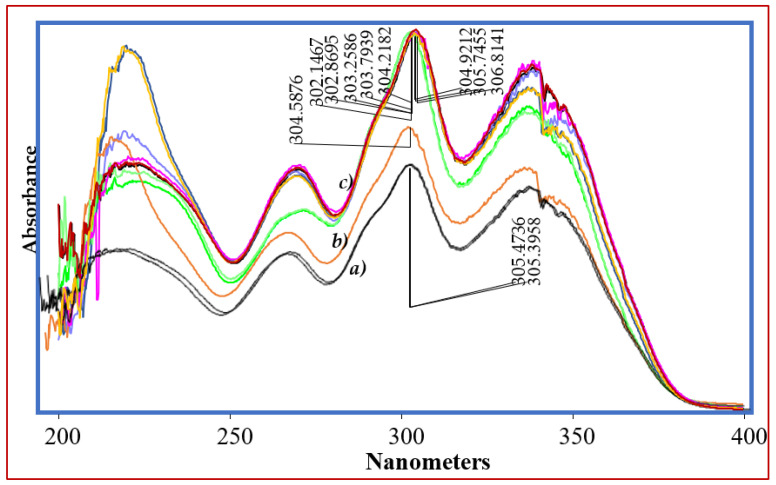
UV absorbance spectrum of VEL, IMPs and DPs.

**Table 1 antibiotics-11-00897-t001:** System suitability studies.

Parameters	VEL	IMP1	IMP2	DP-1	DP2	DP3	DP-4	DP-5	DP-6	DP-7	DP-8
Peak Area (A) mAs	1653.94 ± 25.8	123.51 ± 13.713	112.54 ± 11.9	197.36 ± 18.3	221.76 ± 23.2	245.7 ± 16.7	197.3 ± 19.7	178.9 ± 12.6	213.5 ± 13.5	195.6 ± 16.4	237.9 ± 11.3
Relative standard deviation (RSD)	0.44%	9.41%	9.28%	7.81%	6.98%	5.74%	8.4%	6.87%	7.44%	8.15%	5.41%
Retention Time (tR)	8.5 ± 0.1	8.91 ± 0.1	8.3 ± 0.11	3.1 ± 0.1	7.6 ± 0.1	5.7 ± 0.1	10.4 ± 0.1	11.1 ± 0.1	9.7 ± 0.1	8.3 ± 0.1	2.7 ± 0.1
Relative Retention Time (tRR)	-	1.11375	1.0375	0.3875	0.95	0.7125	1.3	1.3875	1.2125	1.0375	0.3375
Theoretical Plates (N)	14,456	3403	4412	5256	7642	6323	3423	3672	4217	4822	3201
Symmetry Factor (AS)	1.021	0.841	0.907	1.07	0.922	0.925	0.923	0.976	1.02	0.921	0.916
Retention factor K’	8.96	9.93	8.98	4.45	7.55	5.89	9.12	10.74	8.91	7.84	2.11

**Table 2 antibiotics-11-00897-t002:** Results of Recovery and Accuracy.

Analyte	Conc. Level	20%	40%	60%	80%	100%	120%
**VEL**	Recovery%	99.11 ± 0.35	99.45 ± 0.21	101 ± 0.23	101.16 ± 0.46	99.87 ± 0.41	100.61 ± 0.77
	RSD%	0.24	0.35	0.24	0.45	0.39	0.61
**IMP1**	Recovery%	85.12 ± 3.25	89.1 ± 2.25	87.41 ± 2.64	81.16 ± 4.54	89.75 ± 3.63	89.14 ± 5.61
	RSD%	3.31	2.78	2.67	5.12	4.13	6.12
**IMP2**	Recovery%	83.15 ± 4.35	91.3 ± 3.66	87.33 ± 4.12	87.17 ± 3.76	87.41 ± 3.15	92.15 ± 4.71
	RSD%	4.39	4.21	4.75	4.16	3.41	5.02
**DP1**	Recovery%	84.09 ± 3.27	86.17 ± 6.28	83.81 ± 6.15	92.53 ± 4.12	93.56 ± 4.75	95.51 ± 3.45
	RSD%	4.12	7..02	7.21	5.06	4.95	4.17
**DP2**	Recovery%	86.34 ± 5.11	87.69 ± 3.59	91.12 ± 3.86	92.08 ± 6.65	93.86 ± 2.84	94.21 ± 1.76
	RSD%	6.15	3.96	4.09	7.31	3.11	2.92
**DP3**	Recovery%	84.12 ± 6.12	89.22 ± 4.15	92.42 ± 2.86	94.45 ± 6.12	94.86 ± 1.31	96.39 ± 1.87
	RSD%	7.29	5.34	3.03	7.15	1.45	2.11
**DP4**	Recovery%	82.15 ± 7.23	98.44 ± 6.35	90.27 ± 3.97	92.18 ± 5.1	94.56 ± 2.11	93.15 ± 1.64
	RSD%	8.21	7.38	4.10	5.52	2.24	1.71
**DP5**	Recovery%	84.1 ± 6.51	82.17 ± 3.21	93.37 ± 3.26	95.62 ± 4.45	93.31 ± 1.33	92.21 ± 1.29
	RSD%	7.92	3.97	3.31	5.12	1.95	1.86
**DP6**	Recovery%	85.94 ± 8.11	91.23 ± 8.12	94.72 ± 2.51	94.68 ± 1.68	96.11 ± 3.14	94.56 ± 1.59
	RSD%	9.12	8.07	1.75	2.17	4.09	2.22
**DP7**	Recovery%	81.23 ± 6.44	83.54 ± 5.34	83.17 ± 1.18	89.6 ± 1.68	86.25 ± 1.35	81.97 ± 3.67
	RSD%	7.31	6.24	1.23	1.91	1.78	4.02
**DP8**	Recovery%	86.31 ± 7.53	94.39 ± 8.12	91.56 ± 5.41	91.37 ± 4.75	93.34 ± 2.79	95.19 ± 3.34
	RSD%	8.22	8.41	5.60	5.13	3.19	3.46

**Table 3 antibiotics-11-00897-t003:** Precision and linearity studies.

Parameters	VEL	IMP1	IMP2	DP-1	DP2	DP3	DP-4	DP-5	DP-6	DP-7	DP-8
**Precision**	±0.22	±9.12	±8.71	±2.71	±3.24	±6.34	±5.96	±6.23	±5.17	±7.11	±6.32
**Robustness**	±0.76	±8.45	±6.39	±3.86	±6.19	±4.37	±7.34	±5.14	±4.19	±6.51	±6.84
**Slope**	74.69	876.4	655.9	74.5	653.9	899.6	742.7	654.5	955.9	1012	844
**Correlation r**	0.9999	0.9982	0.9987	0.9991	0.9993	0.9924	0.9996	0.9985	0.9994	0.9997	0.9993
**Intercept**	−2.962	−1.073	−7.026	−7.309	−6.541	−15.800	−8.116	−6.481	−6.995	−6.48159	−6.99547
**LOD (µg mL^−1^)**	0.016	0.003	0.017	0.09	0.021	0.016	0.015	0.020	0.016	0.013	0.019
**LOQ (µg mL^−1^)**	0.05	0.010	0.053	0.27	0.064	0.05	0.044	0.06	0.05	0.05	0.05

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
