# Peer review of "Forced Degradation Studies and Development and Validation of HPLC-UV Method for the Analysis of Velpatasvir Copovidone Solid Dispersion"

_antibiotics, 2022, doi:10.3390/antibiotics11070897_

Round 1

Reviewer 1 Report

First of all, I do not feel that the proposed manuscript is sufficient for this particular special issue -  Discovery, and Development of Novel Antibacterial Agents. 

I cannot find any justification to publish it in this issue.

 Authors focus on analytical methods, so it is out of drug discovery and development scope. It can be accepted if they describe the new process our new information about forces degradation of very processing drug candidates. Still, they analyzed the well-known drug substance already on the market. 

Some minor issues should be improved, like figure 1,2, better justification of the study, and why some conditions are essential for evaluation.  

Summarizing, this manuscript in my opinion it does not fit the subject of the journal, and certainly not the subject of the special issue.

Author Response

Comments and Suggestions for Authors

 Reviewer #1: Reviewer's Comments

  1. Comments: First of all, I do not feel that the proposed manuscript is sufficient for this particular special issue - Discovery, and Development of Novel Antibacterial Agents. 

Author’s response:  The forced degradation studies and development of analytical method is presented in the manuscript for the Copovidone dispersion of Velpatasvir (a novel antiviral drug approved by USFDA in June 2016 for the treatment of Hepatitis C). The determination of degradation products and process impurities is integral part of drug development and regulatory requirements. The studies presented in manuscript are not previously published and closely relevant to the scope of special issue Discovery, and Development of Novel Antibacterial Agents.

  1. Comments: I cannot find any justification to publish it in this issue.

Author’s response:  Velpatasvir is a novel antiviral drug and very recently developed for the treatment of Hepatitis C all major genotypes.  For the development of drug product, it is necessary to limit the level of process impurities and degradation products in pharmaceutical formulation and there are strict guidelines [10, 11].  Forced degradation studies are performed to develop a stability indicating analytical method for quantification of degradation products and impurities in drug product.  As there is no studies reported till date to describe the forced degradation and development of stability indicating HPLC-UV method for the determination of impurities and degradation products in Velpatasvir Copovidone dispersion.   Therefore the studies described in the manuscript is a part of drug development and closely relevant to the scope of special issue Discovery, and Development of Novel Antibacterial Agents.

  1. Comments: Authors focus on analytical methods, so it is out of drug discovery and development scope. It can be accepted if they describe the new process our new information about forces degradation of very processing drug candidates. Still, they analyzed the well-known drug substance already on the market.

Author’s response:  As described in the manuscript, to overcome the solubility problem of pure drug substance in the formulations, the drug substance is converted to 50 % w/w dispersion with Copovidone after physical treatment. After any treatment with pure drug substance or converting it into dispersion (binary mixture), separate forced degradation studies are required to evaluate the interaction of binary component with drug substance or degradation products [27, 28]. The present study is completely new and performed on the Copovidone dispersion of novel drug velpatasvir instead of pure drug substance (velpatasvir) and there is no studies available on drug substance in the form of Copovidone dispersion. Similarly the stability indicating analytical method described in the manuscript is validated for determination of impurities and degradation products in Velpatasvir Copovidone dispersion instead of pure drug substance already available in the market.  

  1. Comments: Some minor issues should be improved, like figure 1, 2, better justification of the study, and why some conditions are essential for evaluation.

Author’s response:  Figure-1 describe the process impurities of velpatasvir and figure-2 degradation products. To evaluate the interaction of pure drug substance, impurity or degradation product with a binary component like Copovidone in the dispersion, it is necessary to perform the forced degradation studies in acidic, alkaline and oxidative conditions already described by USFDA guidelines [27, 28]. The forced degradation studies are performed according to the guidelines for the Copovidone dispersion of velpatasvir and the interaction of pure drug substance, impurities and degradation products are studied on HPLC using reference standards of drug substance, impurity or degradation products.

  1. Comments: Summarizing, this manuscript in my opinion it does not fit the subject of the journal, and certainly not the subject of the special issue.

Author’s response: As described above the determination of degradation products and process impurities is integral part of drug development and regulatory requirements. The studies presented in manuscript are performed on a novel antiviral drug (antimicrobial agent) that fit to the subject of the journal antibiotic and similarly closely relevant to the scope of special issue Discovery, and Development of Novel Antibacterial Agents

Reviewer 2 Report

1.   In the figure caption, VEL, IMPs and DPs are mentioned. However, there are six spectra present in figure 4. The authors should revise figure 4 appropriately.

2.   In line number 151, the authors mentioned that the optimum wavelength is 295 nm, and in figure 4, it is marked as 303.25, 303.79 etc. Clarify this.

3.   Mention the IMPs present in the reference standards in the materials and chemicals section.

4.   In figure 3, the labels are not visible in the VEL composite reference standard chromatogram. Moreover, mark the DPs peak in other chromatograms for a better understanding.

5.   The authors are required to include the chromatogram of VEL-CSD in figure 5.

6.   Figure 6 is not cited in the text. If appropriate  the authors can merge Figure 5 and 6. Change VEL-CSDD to VEL-CSD

Author Response

Comments and Suggestions for Authors

Reviewer #2: Reviewer's Comments

  1. Comments: In the figure caption, VEL, IMPs and DPs are mentioned. However, there are six spectra present in figure 4. The authors should revise figure 4 appropriately.

Author’s response: Figure 4 revised, the UV absorbance spectrum of VEL and Process impurities are added.  

  1. Comments: In line number 151, the authors mentioned that the optimum wavelength is 295 nm, and in figure 4, it is marked as 303.25, 303.79 etc. Clarify this.

Author’s response: As evident from Figure-4, the maximum absorbance of velpatasvir, process impurities and degradation products are 305 ± 3 nm. The wavelength 295 nm was corrected accordingly in the line number 151.

  1. Comments: Mention the IMPs present in the reference standards in the materials and chemicals section.

Author’s response: Reference standards of IMPs i.e. IMP1 (C47H52N8O6) and IMP2 (C39H45N7O5), DPS i.e. DP1 (C42H43N7O5), DP2 (C39H45N7O5), DP3 (C32H34N6O2), DP4 (C47H52N8O6), DP5 (C34H34N6O6), DP6 (C32H32N6O6), DP7 (C17H24N4O4) and DP8 (C16H20N2O4) all purity < 84.0 % was provided by Nantong Chanyoo Pharmatech Co. Ltd (China). The material and chemical section is revised accordingly.

  1. Comments: In figure 3, the labels are not visible in the VEL composite reference standard chromatogram. Moreover, mark the DPs peak in other chromatograms for a better understanding.

Author’s response: Figure- 3 revised and the labels of all peaks are corrected. All peaks of DPs in other chromatograms are labelled accordingly. The peaks given in Figure 5 and 6 was also labelled and revised accordingly.

  1. Comments The authors are required to include the chromatogram of VEL-CSD in figure 5.

Author’s response: figure-5 revised, chromatogram for velpatasvir Copovidone dispersion added in figure-5 and labelled accordingly.

  1. Comments Figure 6 is not cited in the text. If appropriate the authors can merge Figure 5 and 6. Change VEL-CSDD to VEL-CSD.

Author’s response: Figure-5 and 6 presents the chromatograms at different conditions, i.e. figure-5 represent the chromatograms for photolytic and oxidative condition and the figure-6 represents the chromatograms for hydrolysis studies therefore both are kept separated. The ligand of figure-6 is corrected and the figure is cited in the manuscript in as appropriated. 

Round 2

Reviewer 1 Report

The authors have revised and satisfied the reviewer comments or suggestions; so I can recoemded it for publication.

Reviewer 2 Report

This manuscript can be accepted for publication in present form.